# FROM STATIC TO DYNAMIC: INFERRING PROTEIN DYNAMICS FROM STRUCTURE AND LANGUAGE EMBEDDINGS

**Arthur Monnier**,* **Pengkang Guo**,* **Bruno Correia, Pierre Vandergheynst**
École Polytechnique Fédérale de Lausanne
Lausanne, Switzerland
{arthur.monnier,pengkang.guo,bruno.correia,pierre.vandergheynst}@epfl.ch

## ABSTRACT

The dynamic behavior of proteins is essential for biological functions such as enzyme catalysis, transmembrane transport, and signal transduction, but its characterization conventionally relies on expensive Molecular Dynamics (MD) simulations. To address this challenge, we develop ResAxial, a deep learning architecture that combines a Residual U-Net with Axial Attention to predict dynamic properties of proteins. We evaluate our model across three input regimes: structure-only (using pairwise distance matrices alone), sequence-only (using ESM-2 embeddings alone), and combined (geometry with sequence embeddings). When combining both modalities with joint supervision on Root Mean Square Fluctuation (RMSF) and correlations, ResAxial achieves strong performance with Correlation PCC=0.959 and RMSF PCC=0.917.

## 1 INTRODUCTION

Proteins are not static entities but dynamic molecules whose motions are crucial for their biological function (Nam & Wolf-Watz, 2023). Understanding these motions requires computing protein dynamic properties and performing dynamic analysis, which generally depend on computationally intensive MD simulations (Hollingsworth & Dror, 2018). Recent machine learning approaches have demonstrated that dynamic properties can be directly predicted from static features, bypassing the need for MD simulations.

Inspired by image-to-image translation tasks in computer vision (Isola et al., 2018) and the success of treating protein residues as 2D grids for spatial feature extraction (Wang et al., 2017), we propose ResAxial, a hybrid architecture combining Residual U-Net with Axial Attention for predicting protein dynamic properties. ResAxial is designed to capture both short-range spatial patterns and long-range dependencies, which are essential for modeling protein dynamics that arise from local contact interactions and distal coupling effects. We investigate how geometric and semantic features contribute to predicting dynamic properties by evaluating ResAxial across three input regimes: structure-only, sequence-only, and combined. We make the following contributions:

1. **Architecture design**: We benchmark convolutional and attention-based architectures and show that a hybrid Residual U-Net + Axial Attention model outperforms both standard U-Nets and pure Transformers on the ATLAS benchmark.

2. **Joint supervision**: Jointly training on RMSF and Correlation improves RMSF prediction by 12%, exploiting their biophysical relationship.

3. **State-of-the-art performance**: When combining structure and sequence modalities, ResAxial achieves an RMSF Pearson Correlation of 0.917, outperforming prior baselines.

---

*Equal contribution.

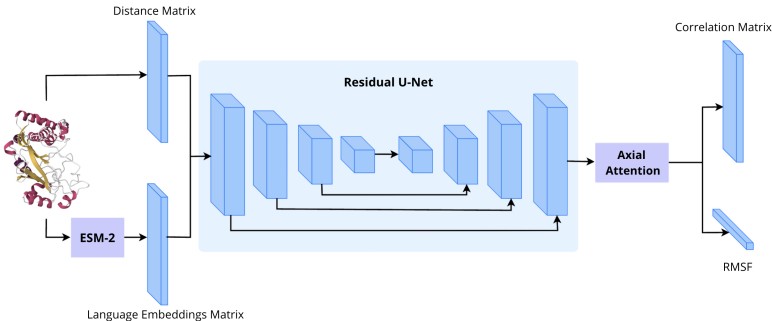

Figure 1: **The ResAxial Architecture.** Our framework processes protein structure (distance matrices) and sequence (ESM-2 embeddings) as a dense image-to-image translation task. A Residual U-Net extracts multi-scale features, followed by an Axial Attention block to predict the inter-residue Correlation matrix and per-residue RMSF profiles.

## 2  RELATED WORK

MD simulations have long been the standard approach for studying protein motions. However, they are computationally expensive and time-consuming, making them challenging for large-scale studies or machine learning pipelines that require extensive training data. Earlier computational methods, such as Normal Mode Analysis (NMA), offer faster alternatives by leveraging simplified physical models; tools like ProDy (Bakan et al., 2011) provide efficient implementations for estimating protein flexibility from coarse-grained representations.

More recently, deep learning methods have been developed to directly predict dynamic properties from static molecular representations. FlexPert (Kouba et al., 2025) employs Graph Neural Networks (GNNs) conditioned on protein language model embeddings to predict RMSF. SeqDance (Hou et al., 2026) relies solely on protein language models to predict relative flexibility. Viliuga et al. (2025) introduced BackFlip, an equivariant neural network that predicts per-residue flexibility from backbone structure alone.

In parallel, work on protein graph representations has explored fusing static and dynamic information. Guo et al. (2025) demonstrated that integrating static distance matrices with MD-derived correlation patterns into heterogeneous graph representations improves performance on downstream tasks, suggesting that static structural features and dynamic correlations provide complementary signals for protein modeling.

## 3  METHODS

### 3.1  RESAXIAL ARCHITECTURE

As shown in Figure 1, we treat protein dynamic property prediction as a dense prediction task. Both the input features and target outputs are naturally represented as 2D maps of dimension $\mathbb{R}^{L \times L \times C}$, where $L$ is the sequence length and $C$ denotes the number of channels. Our proposed model, ResAxial, combines two complementary architectural components:

1. **Residual U-Net**: A symmetric encoder-decoder structure (He et al., 2015) that downsamples spatial dimensions to capture multi-scale features. We use attention gates at skip connections to focus on the relevant spatial regions.

2. **Axial Attention**: To capture global long-range dependencies without the high computational cost of full self-attention, we append a stack of Axial Transformer blocks (Ho et al., 2019; Jumper et al., 2021) to the U-Net final feature map. This factorizes attention into row-wise and column-wise operations, maintaining a global receptive field. For a feature map $H \in \mathbb{R}^{N \times N \times d}$ at index $(i, j)$, where $N$ is the padded sequence length and $d$ is the

feature dimension, the axial update is defined as:

$$y_{i,j} = \text{Attn}_{\text{row}}(H_{i,:}) + \text{Attn}_{\text{col}}(H_{:,j})$$

## 3.2 INPUT FEATURES

We evaluate the model using three feature sets:

1. **Structure-Only (Distance Matrices)**: Structural information is encoded through a distance matrix computed from C$\alpha$ atom coordinates. This gives an $L \times L$ matrix where each entry $D_{i,j}$ represents the Euclidean distance between residues $i$ and $j$. We normalize the distance matrices to the range $[0, 1]$ using min-max normalization per protein.

2. **Sequence-Only (ESM-2 Embeddings)**: For sequence representation, we use ESM-2 (Lin et al., 2022), a protein language model pre-trained on millions of protein sequences. We extract embeddings from the penultimate layer of the ESM-2 model, yielding an $L \times D$ tensor, and $D$ is the embedding dimension. We use the "esm2_t33_650M_UR50D" variant with embedding dimension 1280. To construct the pairwise representation required by ResAxial, for each pair of residues $(i, j)$, we sum their embeddings: $X_{ij} = v_i + v_j$, resulting in an $L \times L \times D$ tensor.

3. **Combined (Structure + Sequence)**: To integrate both modalities, we generate a pairwise representation by summing the projected sequence embeddings and concatenating the result with the structural features. For each pair of residues $(i, j)$, the pairwise feature vector $X_{ij}$ is computed as:

$$X_{ij} = \text{Embed}_{\text{dist}}(e_{ij}) \oplus (\text{Embed}_{\text{seq}}(v_i) + \text{Embed}_{\text{seq}}(v_j))$$

where $\oplus$ denotes concatenation, $v_i$ and $v_j$ are the ESM-2 embeddings for residues $i$ and $j$, and $e_{ij}$ represents the normalized Euclidean distance between their $C_\alpha$ atoms.

## 3.3 TRAINING OBJECTIVE

We train ResAxial to jointly predict both RMSF and Correlation. RMSF quantifies the time-averaged positional fluctuation of each residue:

$$\text{RMSF}_i = \sqrt{\frac{1}{T} \sum_{t=1}^{T} |\mathbf{r}_i^t - \langle \mathbf{r}_i \rangle|^2}$$

where $\mathbf{r}_i^t$ is the position of residue $i$ at frame $t$, and $\langle \mathbf{r}_i \rangle$ is its time-averaged position. The inter-residue Correlation matrix captures the coupling between residue motions:

$$C_{ij} = \frac{1}{T} \sum_{t=1}^{T} \frac{\Delta \mathbf{r}_i^t \cdot \Delta \mathbf{r}_j^t}{|\Delta \mathbf{r}_i^t||\Delta \mathbf{r}_j^t|}$$

where $\Delta \mathbf{r}_i^t$ represents the displacement vector of residue $i$ at frame $t$, and $T$ is the total number of frames. These two properties are related aspects of protein dynamics: RMSF measures local residue flexibility, while Correlation quantifies the coupling between residue motions. We use a joint objective $\mathcal{L} = \mathcal{L}_{\text{RMSF}} + \mathcal{L}_{\text{Corr}}$, where both terms use Mean Squared Error (MSE). We hypothesize that these related properties may provide mutual training signals that improve performance on both tasks.

## 4 EXPERIMENTS AND RESULTS

## 4.1 DATASET

We train and evaluate our models on the ATLAS dataset (Vander Meersche et al., 2023), which contains MD simulation trajectories for a diverse set of proteins. It also contains pre-computed dynamics properties, including per-residue RMSF values. And we calculate the target inter-residue Correlation matrices using PyTraj (Roe & Cheatham III, 2013) and the trajectories provided by ATLAS.

## 4.2 ARCHITECTURE DESIGN

We train all candidate models using the joint objective on both RMSF and Correlation prediction. All experiments use the Pearson Correlation Coefficient (PCC) and Mean Absolute Error (MAE) as evaluation metrics. We use the same topology-based data splits as FlexPert (Kouba et al., 2025).

Table 1: **Architecture comparison.** ResAxial combines local convolution with global attention to achieve the best performance on both RMSF and Correlation prediction.

| Model | Correlation[1] | | RMSF[2] | |
|---|---|---|---|---|
| | PCC[3](↑) | MAE[4](↓) | PCC[3](↑) | MAE[4](↓) |
| CNN Baseline | 0.867 | 0.136 | 0.781 | 0.104 |
| Res-Unet | 0.896 | 0.123 | 0.848 | 0.074 |
| Res-Unet with Attention | 0.890 | 0.126 | 0.859 | 0.075 |
| Axial Transformer | 0.865 | 0.133 | 0.771 | 0.087 |
| **ResAxial** | **0.959** | **0.075** | **0.917** | **0.047** |

[1] Inter-residue Correlation Matrix
[2] Root Mean Square Fluctuation
[3] Pearson Correlation Coefficient
[4] Mean Absolute Error

Table 1 shows that combining local convolution (U-Net) with global attention (Axial) significantly outperforms both approaches independently. ResAxial achieves substantial improvements over the CNN baseline, with Correlation PCC increasing from 0.867 to 0.959 and RMSF PCC from 0.781 to 0.917.

We further compare ResAxial against published baselines on RMSF prediction (Table 2). ResAxial outperforms both BackFlip and FlexPert, achieving a PCC of 0.917 compared to 0.80 and 0.83 respectively.

Table 2: **Comparison against state-of-the-art baselines.** ResAxial demonstrates a significant improvement in RMSF PCC over existing approaches.

| Model | RMSF PCC (↑) |
|---|---|
| BackFlip (Viliuga et al., 2025) | 0.80 |
| FlexPert (Kouba et al., 2025) | 0.83 |
| **ResAxial (Ours)** | **0.917** |

## 4.3 SYNERGY OF JOINT SUPERVISION

We investigate whether joint supervision on RMSF and Correlation improves performance compared to training on each task independently. We compare the ResAxial model trained with single-task objectives versus the joint objective.

Table 3: **Impact of training objectives on ResAxial performance.** Joint training improves RMSF PCC from 0.810 to 0.917 and also slightly improves Correlation performance.

| Training Objective | Correlation | | RMSF | |
|---|---|---|---|---|
| | PCC (↑) | MAE (↓) | PCC (↑) | MAE (↓) |
| RMSF-Only | – | – | 0.810 | 0.078 |
| Correlation-Only | 0.958 | 0.076 | – | – |
| Joint Training | **0.959** | **0.075** | **0.917** | **0.047** |

As shown in Table 3, training on RMSF alone yields an RMSF PCC of 0.810. However, adding Correlation supervision boosts RMSF performance to 0.917 (a 13% relative improvement). Correlation

performance also improves slightly (MAE decreases from 0.076 to 0.075 while PCC increases from 0.958 to 0.959), confirming mutual benefits from joint training. This finding suggests that jointly training on biophysically related properties may be a broadly applicable strategy for protein modeling.

## 4.4 INPUT MODALITY ANALYSIS

To understand how different input modalities contribute to predicting dynamic properties, we evaluate ResAxial across the three input regimes.

Table 4: **Impact of input feature regimes on Correlation and RMSF prediction.** Structure is the dominant factor, but sequence embeddings alone achieve reasonable performance.

| | | Correlation | | RMSF | |
|---|---|---|---|---|---|
| **Regime** | **Input Features** | PCC ($\uparrow$) | MAE ($\downarrow$) | PCC ($\uparrow$) | MAE ($\downarrow$) |
| Sequence-Only | ESM-2 Embeddings | 0.737 | 0.181 | 0.741 | 0.087 |
| Structure-Only | Distance Matrix | 0.944 | 0.082 | 0.886 | 0.052 |
| Combined | Distance Matrix + ESM-2 | **0.959** | **0.075** | **0.917** | **0.047** |

As shown in Table 4, structure is the dominant factor for accurate prediction, with the Structure-Only regime achieving RMSF PCC of 0.886. However, sequence embeddings alone still recover substantial predictive signal (RMSF PCC=0.741), demonstrating that ESM-2 embeddings contain rich latent information about protein dynamics. The Combined regime achieves the best performance, with both modalities providing complementary information.

## 5 CASE STUDY

### 5.1 CORRELATION MATRICES: CAPTURING MOTION CORRELATIONS

As shown in Figure 2, the Combined regime recovers the global correlation pattern with high fidelity, whereas the Sequence-Only regime produces a blurred prediction that fails to resolve fine-grained, off-diagonal coupling. This gap suggests that while ESM-2 embeddings contain enough information for identifying coarse-grained domain movements, they lack the explicit 3D spatial coordinates needed to capture specific fine-grained correlations.

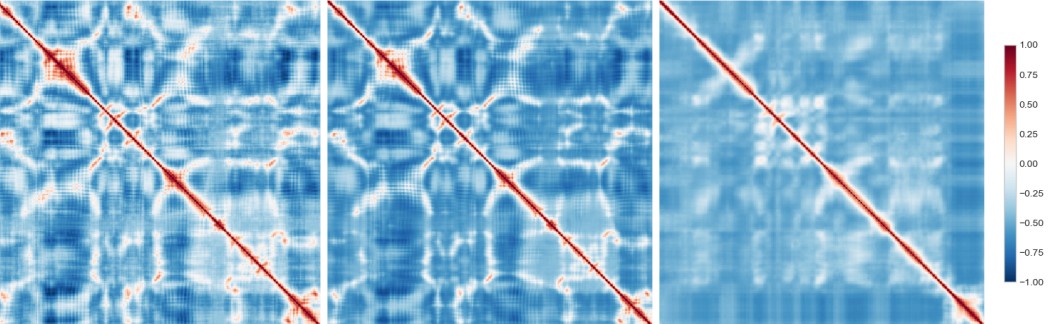

Figure 2: **Comparison of Correlation Matrix prediction (PDB: 1DRV).** (Left) Ground Truth from MD simulations. (Middle) Combined regime. (Right) Sequence-Only regime.

### 5.2 RMSF PROFILES: IDENTIFYING FLEXIBLE REGIONS

The RMSF profiles in Figure 3 demonstrate that while the Sequence-Only regime correctly identifies the locations of major flexible loops, it underestimates their magnitude. The Combined regime captures the fluctuations with much higher precision.

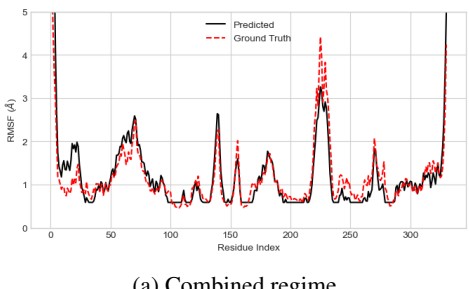
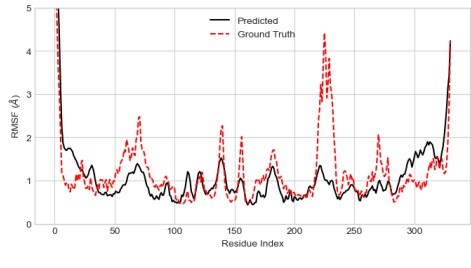

(a) Combined regime         (b) Sequence-Only regime

Figure 3: **Comparison of RMSF Profiles (PDB: 1DRV).**

## 6 CONCLUSION

Machine learning offers a faster alternative to MD simulations for characterizing protein dynamics. In this work, we present ResAxial, a hybrid Residual U-Net with Axial Attention architecture for predicting protein dynamic properties. ResAxial achieves state-of-the-art performance (Correlation PCC=0.959, RMSF PCC=0.917) when combining geometric and semantic features with joint supervision on RMSF and Correlation. Notably, joint training substantially improves RMSF prediction, exploiting the biophysical relationship between these properties. Our evaluation across input regimes shows that while structural geometry is dominant, sequence embeddings alone capture substantial dynamic information, indicating that protein language models contain embedded knowledge of protein dynamics.

### MEANINGFULNESS STATEMENT

A meaningful representation of proteins must capture their dynamic behavior, not just static structure. Our work demonstrates that (1) jointly modeling multiple dynamic properties—RMSF and correlations—enables learning richer and better representations, and (2) protein language models encode latent dynamic information of proteins. By translating static inputs into explicit motion profiles, our ResAxial framework provides functionally relevant representations that bridge structure and dynamics.

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
