# OpenReview forum: "From Static to Dynamic: Inferring Protein Dynamics from Structure and Language Embeddings"
_ICLR.cc/2026/Workshop/LMRL — ICLR 2026 Workshop LMRL Poster_

### Official Review · Reviewer_vXWH · 2026-02-22
**The authors propose ResAxial that models protein dynamics by capturing both short range spatial patterns and long range dependencies. They combine a residual U-neT and axial attention architecture and show the approach outperforms on ATLAS benchmarks.**

**Rating:** 4
**Confidence:** 3

**Review:**

The motivation behind the approach is sound since MD simulations are costly and time consuming and getting fine grained MD data at scale is a challenge. The model is inspired by vision translation tasks where the authors use a Residual U-Net with Axial Attention to predict dynamic properties. This is indeed a clever idea with the latter getting short and long range interactions in addition to avoiding the cost of full self attention. Methods section is easy to follow and the results on architecture comparison shows promise (Table 1). Below are some comments

1. The choice of using sequence only models for getting sequence representation might not be enough since there are several papers that have come up since ESM-2 models that are structure pLMs (SaProt, ProstT5) and it would be nice to do some ablations on different type of embeddings to see which component of the model contributes to the highest gains.

2. Similar to the above comment while summing the sequence embeddings is a good start there can be other ways of combining intra and inter modality (for e.g. cross attention between sequence and structure like the LMDesign paper has done). This would also reveal the benchmark and best way to combine modalities.

3. While Atlas is a good starting point, it is important to see how much the model can generalize to unseen proteins by fold, families. And also how much sensitive is the model to different simulation time of MD.

If the authors could address the above comments, it would make their claim much stronger.

---

### Official Review · Reviewer_QCFg · 2026-02-23
**New method for predicting RMSF and residue position pairwise correlation from static structures but insufficient comparison with the state-of-the-art**

**Rating:** 5
**Confidence:** 3

**Review:**

## Overview
This paper proposes a novel deep learning model to predict protein dynamic properties (RMSF and inter-residue correlations) from a pairwise inter-residue distance matrix and protein language model embeddings. It demonstrates improved results compared to previous work in RMSF prediction. The paper also proposes an ablation study, which demonstrates that jointly predicting RMSF and correlations improves RMSF prediction performance, and that jointly using sequence and structure information improves the performance compared to single-modality approaches.

## General appreciation
This model addresses prediction of dynamic properties of proteins (inter-residue correlation, RMSF), which holds potential to overcome MD heavy computational requirements. The model shows improved performance over 2025 models for RMSF prediction and provides a rigorous ablation study. However, it does not benchmark against other methods for the second task, correlation prediction, thus preventing to position this work's performance for correlation prediction. Besides, the writing lacks technicalities and interpretation of the results of the ablation study.

## Pros and cons
**Pros:**
* Inferring protein dynamics from a static structure holds potential, since it is a computationally lighter way of predicting correlations and RMSFs than Molecular dynamics, and this information has been shown to improve performance on downstream property prediction tasks
* The results are above other models (BackFlip, FlexPert) for RMSF prediction
* The ablation study is extensive, and the fact that the joint use of protein language model embeddings and structures outperforms the use of a single modality is a nice finding

**Cons:**
* The related work overlooks RocketSHP [1], which predicts RMSF and residue pairwise generalized correlation coefficients from static structures
* As a consequence, the performance of ResAxial on correlation prediction is not benchmarked against other models, such as RocketSHP [1] or MD-derived values. This does not allow us to determine whether ResAxial is a state-of-the-art method for correlation prediction
* The paper lacks important technical details (the model hyperparameters are not specified and the splitting strategy is not discussed whereas it is a critical consideration in structural biology because of potential non-independence and high similarity between proteins)
* The ablation study, though extensive, is too descriptive: the results are not interpreted

[1] Sledzieski, Samuel, and Sonya Hanson. "Rocketshp: Ultra-fast proteome-scale prediction of protein dynamics." bioRxiv (2025): 2025-06.

---

### Official Review · Reviewer_6wF9 · 2026-02-25
**From Static to Dynamic: Inferring Protein Dynamics from Structure and Language Embeddings - Review**

**Rating:** 6
**Confidence:** 3

**Review:**

## Summary



This paper presents ResAxial, a hybrid Residual U-Net with Axial Attention architecture for predicting protein dynamic properties (RMSF and inter-residue correlations) from static features. The model is evaluated across three input regimes: structure-only (pairwise distance matrices), sequence-only (ESM-2 embeddings), and combined. Joint supervision on RMSF and correlation is shown to substantially improve RMSF prediction (+13% relative PCC). On the ATLAS benchmark, the combined regime achieves Correlation PCC=0.959 and RMSF PCC=0.917, outperforming BackFlip and FlexPert baselines.



## Strong Points



**S1. Comprehensive ablations (strong).** The paper provides systematic ablations across three axes: architecture (Table 1), training objective (Table 3), and input modality (Table 4). Each table isolates a specific design choice. Table 4 in particular is informative -- it shows that ESM-2 embeddings alone recover substantial dynamic signal (RMSF PCC=0.741), that structure dominates (PCC=0.886), and that combining them yields a clear additive benefit (PCC=0.917).



**S2. Clear writing (moderate).** The paper is concise and well-structured. The problem, method, and experiments are laid out cleanly, and the figure/table quality is good for a workshop paper.



**S3. Benchmarking (moderate).** Comparisons against a CNN baseline (Table 1) and two published methods -- BackFlip and FlexPert (Table 2) -- provide useful context, though the comparison has caveats (see W1).



## Weak Points



**W1. Overstated joint training claims (moderate).** The paper claims joint training provides "mutual benefits" (Section 4.3), but the benefit is almost entirely one-directional. RMSF PCC improves substantially (0.810 → 0.917), but correlation PCC moves from 0.958 to 0.959 -- indistinguishable from noise without error bars. The more precise conclusion is that correlation supervision provides a strong auxiliary signal for RMSF, not that there is a mutual benefit. There is also a minor inconsistency: Contribution 2 says "12%" while Section 4.3 says "13%."




**W2. Distance normalization (moderate).** Structure embeddings use per-protein min-max normalization, which eliminates absolute distance information. The strength of physical interactions between residues depends on absolute distances (e.g., van der Waals, electrostatic), so this choice discards potentially important signal. No ablation on the normalization technique is provided.



**W3. Limited case study (minor).** The case study uses a single protein (PDB: 1DRV). It is difficult to draw general conclusions about when the sequence-only regime succeeds or fails from one example.



## Recommendation



**Decision: Accept**



**Key reasons:**



1. The paper addresses a clear problem -- predicting protein dynamics from static features -- with a well-designed architecture and systematic experiments across modalities, objectives, and architectures.

2. Table 4 provides genuinely informative results: ESM-2 embeddings alone encode substantial dynamic information, and the combined regime shows clear additive benefit over either modality alone.



**Supporting arguments:**



The ablations across input modalities (Table 4) are the strongest contribution. They show that pLM embeddings contain latent dynamic information not fully captured by structure alone, which is a useful insight for the protein ML community. The architectural ablation (Table 1) also convincingly demonstrates the value of combining local convolution with global attention. The weaknesses are concentrated in evaluation rigor (missing error bars, overstated joint training claims) rather than the core methodology, which is appropriate to flag but acceptable for a workshop paper.



## Questions for Authors



1. **Normalization ablation.** The min-max normalization of distance matrices destroys absolute distance information, which encodes interaction strength. Did you ablate this choice (e.g., comparing against log-transform, fixed-range clipping, or unnormalized distances)? If so, what were the results?

2. **Per-region analysis of pLM embeddings.** Did you analyze prediction quality from ESM-2 embeddings across different structural contexts ? This could reveal whether pLM embeddings encode dynamics uniformly or are biased toward specific structural motifs.

3. **Fair baseline comparison.** Can you report the structure-only ResAxial result alongside BackFlip and FlexPert in Table 2 to enable a controlled comparison?



## Additional Feedback



These points are intended to help improve the paper and are not part of the decision assessment.


- **CNN baseline.** CNNs are not the standard architecture for protein structure tasks -- GNNs are more common. Including a GNN baseline in Table 1 would strengthen the architecture ablation.

---

### Meta-Review · Area_Chair_EAF8 · 2026-02-28

**Recommendation:** Accept (Poster)
**Confidence:** 3

**Metareview:**

While the reviewers point out limitations which make it clear that this work should be developed further, I think it is worth discussing at the workshop.

---

### Decision · Program_Chairs · 2026-03-02

**Decision:**

Accept (Poster)

**Comment:**

Please see the meta-review.